# The Mechanisms of GPR55 Receptor Functional Selectivity during Apoptosis and Proliferation Regulation in Cancer Cells

**DOI:** 10.3390/ijms24065524

**Published:** 2023-03-14

**Authors:** Mikhail G. Akimov, Natalia M. Gretskaya, Polina V. Dudina, Galina D. Sherstyanykh, Galina N. Zinchenko, Oksana V. Serova, Ksenia O. Degtyaryova, Igor E. Deyev, Vladimir V. Bezuglov

**Affiliations:** Shemyakin-Ovchinnikov Institute of Bioorganic Chemistry, Russian Academy of Sciences, Miklukho-Maklaya 16/10, 117997 Moscow, Russia; akimovmike@gmail.com (M.G.A.);

**Keywords:** CB1, CB2, GPR18, GPR55-CB2 heterodimers, GPR55 functional selectivity, GPR55 proliferation stimulation, LPI, N-acyl dopamines

## Abstract

GPR55 is a non-canonical cannabinoid receptor, important for cancer proliferation. Depending on the ligand, it induces either cell proliferation or death. The objective of the study was to establish the mechanisms of this multidirectional signaling. Using the CRISPR-Cas9 system, the GPR55, CB1, CB2, and GPR18 receptor knockouts of the MDA-MB-231 line were obtained. After the CB2 receptor knockout, the pro-apoptotic activity of the pro-apoptotic ligand docosahexaenoyl dopamine (DHA-DA) slightly increased, while the pro-proliferative activity of the most active synthetic ligand of the GPR55 receptor (ML-184) completely disappeared. On the original cell line, the stimulatory effect of ML-184 was removed by the CB2 receptor blocker and by GPR55 receptor knockout. Thus, it can be confidently assumed that when proliferation is stimulated with the participation of the GPR55 receptor, a signal is transmitted from the CB2 receptor to the GPR55 receptor due to the formation of a heterodimer. GPR18 was additionally involved in the implementation of the pro-apoptotic effect of DHA-DA, while the CB1 receptor is not involved. In the implementation of the pro-apoptotic action of DHA-DA, the elimination of Gα_13_ led to a decrease in cytotoxicity. The obtained data provide novel details to the mechanism of the pro-proliferative action of GPR55.

## 1. Introduction

GPR55 is a G-protein-coupled receptor of the non-classical cannabinoid receptor family [1]. It is expressed in many organs and tissues of the body [2]. Transcripts of this receptor have been found in the central nervous system (the striatum, hypothalamus, mi-croglia, etc.). This receptor is present in the spleen, bone marrow, platelets, neutrophils, and lymphocytes, which indicates its involvement in the inflammatory response. The GPR55 receptor has been identified in various types of cancer cells (glioma, melanoma, breast, and pancreatic cancer) [3]. 

Initially, it was assumed that its ligands were endocannabinoids (anandamide, 2-arachidonoylglycerol, etc.); however, later, it was found that the active activator of this receptor is alpha-lysophosphatidylinositol (LPI), which is considered the primary natural ligand [4] of the receptor. We demonstrated that another natural ligand for this receptor can be arachidonoyl dopamine, as well as other acyl dopamines [5,6]. Thus, the GPR55 receptor is currently positioned as a multi-ligand receptor.

N-acyl dopamines are a family of bioactive lipids of the endocannabinoid/endovanilloid family [7]. Structurally, they are amides of dopamine and arachidonic (AA-DA), docosahexaenoic (DHA-DA), oleic (Ol-DA), and other fatty acids. Mammals are able to both synthesize and degrade such molecules [7,8]. Their primary receptor target is considered TRPV1; however, recently, it appeared that the GPR55 could also be activated by these molecules [7,9].

The activation of GPR55 by LPI leads to increased cell proliferation; therefore, if an increased expression level of this protein is detected in tumors, this is considered a marker of a poor prognosis for the development of the disease [10,11]. In addition, GPR55 is also involved in other processes. For example, the activation of this receptor protects neurons from excitotoxicity in a model of cultured hippocampal slices [12] and has a protective effect in a neuroinflammation model [13]. LPI, through the activation of GPR55, has a microglia-mediated neuroprotective effect [12] and induces the differentiation of retinal cells [14].

The effect of the GPR55 receptor activation thus highly depends on the ligand, the phenomenon known as “agonist functional selectivity” [15]. As such, LPI induces proliferation [16] or migration [17], while anandamide and acyl dopamines induce apoptosis of cancer cells. The exact mechanism of cell death of the latter process may differ, for exam-ple, in the cells of rat pheochromocytoma PC12, acyl dopamines induced the expression of NO synthase, which in turn generated reactive nitrogen and oxygen species as a by-product [6]. In the case of anandamide in cholangiocarcinoma cells, the death receptor is activated with the participation of GPR55 [18]. It should be noted that GPR55 activation leads to cell death via the apoptosis induction. Depending on the ligand, either the extrinsic [18] or intrinsic [5,9] pathways could be activated.

The interaction of GPR55 with different pathways within the cell also varies depending on the ligand. Thus, in HEK 293 cells transfected with GPR55, LPI, the best-known agonist of this receptor, causes intracellular calcium release, nuclear factor of activated T-cells (NFAT) activation, ERK1/2 phosphorylation, CREB, and NFκB activation. At the same time, treatment with a CB1 receptor antagonist (without LPI) has a similar effect on calcium and CREB, and it is even more effective in the latter case, but suppresses ERK phosphorylation and NFκB activation [19].

The activity of GPR55 ligands depends on the concentration in a non-standard way: there is evidence that low concentrations of anandamide activate the receptor, while high concentrations inhibit it [20].

Literature data on GPR55 intracellular signaling indicate that this receptor, depending on the cell line and conditions, is able to interact with several Gα subunits, namely Gα_13_, Gα_q/11_, Gα_12_, and Gα_12/13_ [21], while the Gα_i1/2_, Gα_i3_, and Gα_s_ subunits do not interact with this receptor [15]. The potential for interaction with several Gα subunits has also been described for other G-protein-coupled receptors, for example, after activation of the LPA4 receptor, neurite retraction mediated by Gα_12/13_ and Rho, calcium mobilization with the participation of Gα_q_, and an increase in cAMP levels with the participation of Gα_s_ [15]. At the same time, in the case of the LPA4 receptor, the activation of different Gα subunits is not a property of the same receptor in different states, but rather is due either to different isoforms of the receptor or to the fact that only a part of the G alpha subunits is present in the cells.

Another peculiar property of the GPR55 signaling is its ability to form heterodimers with cannabinoid receptors CB1 and CB2. The presence of CB1 reduces ERK activation caused by GPR55 ligands, while the presence of GPR55, on the contrary, enhances the activation of this kinase by CB1 ligands [22]. Upon dimerization of GPR55 with CB2, the first receptor suppresses the activity of the second one [23].

The fact that a receptor important for cancer proliferation can be switched to the cell death induction mode just by changing its ligand seems to be of the utmost importance, as it provides a possibility to develop novel therapeutic approaches, which would simultaneously induce the cell death via GPR55 activation and force cancer cells to remove this receptor from their surface, thus removing its pro-proliferative signaling. The mechanism of the agonist functional selectivity of GPR55 is, however, not known. 

We propose two main mechanisms of the effect of the multidirectional action of the ligands of this receptor:Different ligands can induce dimerization or bind to different heterodimers of the GPR55 protein with CB1, CB2, or GPR18 receptors (CB1-GPR55, CB2-GPR55, and GPR18-GPR55) and thereby induce different intracellular signaling cascades.Different ligands can induce the coupling of the GPR55 receptor to different Gαα subunits (Gα_q_, Gα_12_, or Gα_13_), thus inducing opposite consequences for the cell (apoptosis or proliferation).

In this paper, we used CRISPR knockout and siRNA knockdown to compare the cytotoxic effects of the GPR55 ligand DHA-DA in the experimental setting, where the ability of GPR55 to form various heterodimers and activate particular Gα subunits was disrupt-ed. GPR18 and CB1 appeared to function independently of GPR55, while the CB2-GPR55 heterodimer participated in a pro-proliferative activity. In addition, the G α_13_ subunit was more important in the cytotoxicity induction compared with other Gα subunits.

## 2. Results

### 2.1. CB1-GPR55 Interaction

We started our work by testing the existence of some effects of the GPR55 ligands that could be explained by the interaction of CB1 and GPR55 receptors. We hypothesized that, if such an interaction exists, the effect of a functionally selective ligand should be different in a cell line with and without CB1. The model cell line was chosen to be the MDA-MB-231 cell line, as it naturally expresses all cannabinoid receptors [24]. To assess this hypothesis, we first obtained an MDA-MB-231 cell line clone with a CB1 receptor knockout.

The knockout lines were generated using the CRISPR-Cas9 system described by Ran et al. [25]. At the first stage, plasmid constructs based on the PX459 vector (pSpCas9(BB)-2A-Puro) containing sequences of 20 nucleotides complementary to the modified genomic DNA region were obtained. The target sequence was chosen so that it contained the PAM sequence for the SpCas9 on its 3′ end (5′-NGG). To facilitate primary knockouts screening, at the 3’ end the target sequences also contained cleavage sites for restriction enzymes, so that in the case of successful genomic modification one or more nucleotides would be excised, and the restriction site would disappear.

In the case of the CB1 receptor, the control restriction site was PvuII. In clone 3, this restriction site was completely lost, and in clones 4 and 5, the loss was only partial (Appendix A); therefore, clone 3 was chosen for further study.

We compared the cytotoxicity of the DHA-DA on the source MDA-MB-231 cell line and clone 3 (Figure 1, Table 1) after the 24 h incubation. No statistically significant difference was observed, and so we considered the CB1-GPR55 interaction to be insignificant for the GPR55 functional selectivity during the cytotoxicity induction. There were also no substantial effects of the knockout on the caspase activation (Figure 2) and GPR55 ligand ML-184 pro-proliferative action (Figure 3).

### 2.2. CB2-GPR55 Interaction

The next possible receptor pair in the GPR55 functional selectivity was CB2-GPR55. For this receptor pair, we used the same methodology as for CB1. For this receptor, the target sequence contained the PasI restriction site. The analysis of the CB2 gene fragment from the transfected clones revealed the absence of this site (Appendix A), which indicated that the target gene was modified in several clones.

DHA-DA was cytotoxic to both the original cell line and the clone with a knockout CB2 receptor (Figure 4, Table 2). At the same time, for the original cell line the EC50 value was 40.58 µM, and for the knockout clone, it was 33.57 µM, and this difference was statistically significant. According to the sequence analysis of clones, it was in the clone where an increase in DHA-DA cytotoxicity was observed that the expression of the CB2 receptor was disrupted.

Thus, the removal of the CB2 receptor did not lead to a decrease in toxicity and, therefore, the CB2-GPR55 dimer is not involved in this type of activity. However, the increase in DHA-DA cytotoxicity after the removal of the CB2 receptor provides indirect evidence for the interaction of these two G-protein-coupled receptors.

The increase in DHA-DA cytotoxicity after the CB2 receptor knockout could be a result of the disappearance of the pro-proliferative signal transmitted from GPR55 via an interaction with the CB2 receptor. To test this assumption, we next evaluated the activity of the GPR55 agonist ML184 on the original cell line, and the line knocked out at the CB2 receptor (Figure 5). This revealed that the stimulation of the proliferation was no longer observed on knockout, which suggested the likely involvement of the CB2 receptor in the stimulation of the proliferation with the participation of the GPR55 agonists.

The observed stimulation of proliferation and its disappearance in the CB2 knockouts could be interpreted in several ways: a direct interaction of the ligand with the CB2 receptor without the participation of GPR55; a signal transmission in the GPR55-CB2 complex; and damage in the process of knockout of other cell systems that are not related to the direct interaction of CB2 and GPR55.

To clarify the issue, the action of the ML-184 agonist was tested on the original cell line against the background of GPR55 (ML-193) and CB2 (SR 144528) blockers, as well as on the MDA-MB-231 cell lines knocked out for the GPR55 receptor.

To generate the GPR55 knockouts, the same approach as for the CB1 and CB2 receptors was used. The target sequence for the GPR55 receptor was chosen to contain the NcoI restriction site. The restriction of the PCR fragment showed the absence of the NcoI restriction site in the modified DNA fragment. Sequencing confirmed the presence of deletions in the modified gene in different alleles in C12 and F10 cell clones (Appendix A), indicating successful knockout.

The stimulatory effect of ML-184 on the original cell line was removed by the CB2 receptor blocker, but not by the blocker of GPR55 (Figure 6). At the same time, the stimulating effect of ML-184 was practically not manifested on cell lines knockout at the GPR55 receptor (Figure 7).

To additionally verify the hypothesis on the pro-proliferative signaling from DHA-DA via the CB2 receptor, we measured cell proliferation using the BrdU kit and caspase activation using the fluorogenic substrates in the native and CB2 knockout cell line. To avoid the pro-proliferative effects of the lipids from the serum, delipidated serum was used. CB2 knockout prevented DHA-DA pro-proliferative signaling but had no effect on the caspase activation (Figure 8 and Figure 9), thus confirming the hypothesis.

To validate the observed effects, we tested the ability of ML-184 and LPI to stimulate the proliferation of two other cell lines, Mia PaCa-2 and Hep G2, which express GPR55 but lack CB2 [26,27,28,29]. Neither of the substances was able to stimulate the proliferation of these cells (Figure 10), confirming the assumption of the role of the CB2 receptor in the proliferation induction via GPR55.

Thus, it can be safely assumed that the stimulation of proliferation with the participation of the GPR55 receptor results in signal transduction from the CB2 receptor to the GPR55 receptor with the possible formation of a heterodimer.

### 2.3. GPR18-GPR55 Interaction

The next possible receptor pair in the GPR55 functional selectivity was GPR18. For this receptor pair, we used the same methodology as for CB1. The target CRISPR sequence contained the BstAUI restriction site. The analysis of the GPR18 gene fragment from the transfected clones revealed the absence of this site (Appendix A), which indicated that the target gene was modified in clone 5.

DHA-DA was cytotoxic to both the original cell line and the clones with a knockout GPR18 receptor (Figure 11, Table 3). However, for the clones with a disrupted expression of the GPR18 receptor, the EC_50_ values of the substance were significantly higher.

To check for the importance of the possible GPR55-GPR18 interaction during the cytotoxicity induction, we also tested the effect of the GPR18 blocker PSB CB5 on DHA-DA cytotoxicity for the unmodified cell line, and the toxicity of DHA-DA for the GPR55 knockouts (Figure 12, Table 4).

Both the GPR18 blocker and GPR55 knockout led to a decrease in the DHA-DA cytotoxicity. These data together with our previous data on the ability of the GPR55 blocker to reduce DHA-DA cytotoxicity [9] indicate that DHA-DA interacts with each of the receptors and that their dimer, if any, does not play a substantial role in the cytotoxicity induction. GPR18 knockdown led to a slight decrease in caspases 9 and 3 activation for the same DHA-DA concentration, which was in agreement with the reduced cytotoxicity signal (Figure 13). On the other hand, the response to ML-184 did not change (Figure 14), indicating that the GPR55-GPR18 interaction, if any, does not play a significant role in pro-proliferative GPR55 signaling.

### 2.4. G Protein Knockdown Effect

Another explanation for the functional selectivity of the GPR55 receptor could be its interaction with different Gα subunits depending on the ligand. According to this hypothesis, removal of the particular Gα subunit would result in a substantial change in the ligand activity, while the absence of other kinds of Gα subunits would have little or no effect. As far as the viability of the cell could be drastically changed if a Gα subunit is completely lost, and given the absence of selective inhibitors for all of the existing Gα subunits, we used siRNA knockdown to decrease their expression. RT-qPCR was used to confirm the drop of the appropriate Gα mRNA expression (Figure 15).

DHA-DA cytotoxicity significantly differed for the Gα_13_ knockdown (Figure 16, Table 5), indicating the preferential participation of this subunit in the cytotoxicity induction.

As far as the second mode of action for the GPR55 receptor is the induction of proliferation, we evaluated the activity of its known pro-proliferative ligand LPI on the Gα subunit knockdowns (Figure 17). No significant difference from the cells transfected with the scrambled siRNA control was observed, indicating the absence of a preferred role of a particular Gα subunit in the pro-proliferative activity induction via the GPR55 receptor.

## 3. Discussion

The observed ability of the GPR55 receptor to induce both proliferation and cell death in cancer cells depending on the agonist seems to be a very interesting therapeutic opportunity. It potentially allows for the situation when an acquired drug resistance via a decrease in the receptor expression (a typical way to avoid a targeted therapy) would impair cell viability by removing one of the pro-proliferative signaling points. However, to avoid a situation when a poorly designed drug would instead stimulate cell proliferation, an understanding of the functional selectivity mechanism of the GPR55 receptor is required. We found that this phenomenon substantially relies on the GPR55-CB2 receptor interaction, most probably in the form of the heterodimer formation, which could be dependent on the expression level of the Gα_13_ subunits and enhanced by the presence of the GPR18 cannabinoid receptor.

Within the framework of the model system of this study, the participation of the CB1-GPR55 heterodimer in the cell proliferation regulation was not observed, as the knockdown of the CB1 receptor did not change GPR55 ligand DHA-DA cytotoxicity. CB1 and GPR55 receptors are co-expressed and form heteromers in rat and monkey striatum. In the heterologous system, the 1 μM CID1792197 (GPR55 agonist)-induced activation of NFAT was blocked by the selective CB1 antagonist SR141716. The increase in pERK1/2 due to the GPR55 agonist in co-transfected cells was blocked by pretreatment with SR141716, the CB1 antagonist [30]. The observed lack of interaction during the DHA-DA cytotoxicity could be due to the fact that CB1 usually transmits anti-proliferative signals [31], and thus has little effect in the acute setting.

In this study, the interaction of CB2 and GPR55 receptors was found to be important during the pro-proliferative signal transduction from the GPR55 receptor. Without CB2, neither LPI nor the selective GPR55 agonist ML-184 were able to stimulate the proliferation of the cells via the GPR55 receptor. Moreover, the signaling from DHA-DA appeared to have both a cytotoxic and a pro-proliferative component, the latter being masked by the former. The signal was transmitted from GPR55 to CB2, as the pro-proliferative effect was observed for a highly selective GPR55 agonist, and this effect disappeared after CB2 knockdown. CB2 and GPR55 are known to form heteromers in cancer cells. Via these complexes, the agonists and antagonists of one receptor are able to impair the signaling of the partner receptor. At low concentrations, THC (a well-known CB2 agonist) signals through CB2, producing a conceivable activation of ERK1/2 and inhibition of FK-induced cAMP increase. At higher concentrations, THC is able to target GPR55, acting as a receptor antagonist and exerting a cross-antagonism over CB2 through the heteromer, which would result in an attenuation of the CB2-mediated effects on ERK1/2 activation and cAMP production [23]. Whereas heteromerization leads to a reduction in the GPR55-mediated activation of transcription factors (NFAT, NFκB, and cAMP response element), ERK1/2-MAPK activation is potentiated in the presence of CB2 receptors. CB2 receptor-mediated signaling is also affected by co-expression with GPR55 [32]. Our data, however, extend this picture to the area of the GPR55-only agonists, which appear to be highly dependent on CB2 to exert a pro-proliferative action. In the heterologous GPR55 expression models, LPI behaves as a GPR55 agonist [4]. However, based on our data, it could be supposed that this interaction is not always enough to induce proliferation and may require CB2 receptor presence to manifest.

In our experimental setting, a cytotoxic GPR55 agonist DHA-DA exerted its effect via both GPR55 and GPR18. Both the GPR18 blocker and GPR55 knockout led to a decrease in the DHA-DA cytotoxicity but did not prevent it completely. These data agree with the previous research, in which pro-apoptotic GPR55 agonists realized part of their action through the activation of GPR18 [33]. We proposed that heterodimer formation between GPR55 and GPR18 either does not occur or does not play a substantial role in the DHA-DA cytotoxicity induction. GPR18 is known to form heterodimers with CB2, but GPR18-GPR55 heterodimers are not known. CB2-GPR18 heteroreceptor complexes displayed particular functional properties often consisting of negative cross-talk and cross-antagonism (the response of one of the receptors is blocked by a selective antagonist of the partner receptor) [34].

Our experiments on the Gα subunit knockdowns showed that DHA-DA cytotoxicity markedly depended on the Gα_13_ presence, while the stimulation of the proliferation by LPI did not preferentially depend on any particular subunit tested. This role of the Gα_13_ subunit in the GPR55 effects was not described before; however, for other receptors, a similar behavior is known. For example, for the CCKAR receptor, compared with Gα_s_ or Gα_i_ proteins, Gα_q_ coupling increases the binding affinity of CCK-8, consistent with the increased binding activity of isoproterenol against β2AR in the presence of Gα_s_ protein [35]. G protein coupling may allosterically influence ligand binding, and that could be the cause of the observed decrease in DHA-DA cytotoxicity in the absence of Gα_13_. G protein coupling to the β2AR stabilizes a “closed” receptor conformation characterized by restricted access to and egress from the hormone-binding site. In contrast to agonist binding alone, the coupling of a G protein in the absence of an agonist stabilizes large structural changes in a GPCR [36]. However, we cannot exclude the participation of other tested Gα proteins in effects of GPR55 because their inactivation was not complete (and hardly possible within the available experimental framework). Therefore, further investigations are necessary to fully resolve this problem.

Of all the results obtained, the novel data on the role of the CB2-GPR55 heterodimer seem to be the most interesting. These data indicate that, first, a combination of a GPR55 agonist with a CB2 blocker could be an efficient anti-cancer drug combination, and second, that in the tumors positive for both GPR55 and CB2, a CB2 knockout or knockdown could lead to a potential switch of GPR55 from the pro-proliferative to the cytotoxic mode. However, the question remains: What allows some of the GPR55 agonists, and not others, to induce the cytotoxic response? The answer may be that the action mode of GPR55 without heterodimers is cytotoxicity induction and not proliferation, and this is an important area for further research.

## 4. Materials and Methods

### 4.1. Reagents

Isopropanol, MTT, D-glucose, DMSO, RPMI 1640, DMEM, L-glutamine, HCl, Triton X-100, Hank’s salts solution, Versene’s solution, penicillin, streptomycin, amphotericin B, RPMI 1640, DMEM, trypsin, (4,5-dimethylthiazol-2-yl)-2,5-diphenyltetrazolium bromide (MTT), and fetal bovine serum were obtained from PanEco, Moscow, Russia.

Cell lines MDA-MB-231 (HTB-26), Mia PaCa 2 (CRL-1420), Hep G2 (HB-8065), and RPMI-8226 (CCL-155) were purchased from ATCC, Manassas, VA, USA.

Antibodies anti-b-actin, anti-CB2, anti-GPR55, and BrdU cell proliferation assay kit were obtained from Abcam, Cambridge, UK. Anti-mouse IgG antibody was obtained from Santa-Cruz Biotechnology, Dallas, TX, US. siRNA for the Gα_13_, Gα_12_, Gα_q_, and scrambled siRNA control were obtained from Santa-Cruz Biotechnology, Dallas, TX, US.

Pan-caspase inhibitor Z-VAD-FMK (N-benzyloxycarbonyl-Val-Ala-Asp(O-Me) fluoromethyl ketone) and caspase substrates Ac-DEVD-AFC and Ac-LEHD-AFC were obtained from Tocris Bioscience, Bristol, UK.

SR 144028, PSB CB5, ML-184, ML-193, and LPI were obtained from Tocris Bioscience, Bristol, UK. Glycylglycine, acetic acid, MgSO_4,_ EGTA, dithiothreitol, DMSO, Triton X-100, acrylamide, bis-acrylamide, SDS, nitro blue tetrazolium, Tris-Borate-EDTA, agarose, bicinchoninic acid, D-glucose, bovine serum albumin, anti-rabbit IgG antibody, SCP0139, and 5-Bromo-4-chloro-3-indolyl phosphate-toluidine were obtained from Sigma-Aldrich, St. Louis, MO, USA. siRNA, RNAiMax, Advanced DMEM, and DreamTaq master mix were obtained from Thermo Fisher Scientific, Walthon, MA USA. Total RNA Purification kit was obtained from Jena Biosciences, Jena, Germany. MMLV reverse transcription kit and SYBR Green HS master mix was obtained from Evrogen, Moscow, Russia. The purity of all used reagents was 95% or more.

### 4.2. Chemical Synthesis

Dopamine amide of docosahexaenoic acid was synthesized as described previously [37].

### 4.3. Cell Culture

Cells were cultured in DMEM (cell lines MDA-MB-231, Mia PaCa 2, Hep G2) or RPMI-1640 (cell line RPMI-8226) supplemented with 10% FBS, 4 mM L-glutamine, 100 U/mL penicillin, 100 μg/mL streptomycin, and 2.5 μg/mL amphotericin B in 5% CO_2_ and 100% humidity at 37 °C. Cells were subcultured by successive treatment with Versene’s solution and trypsin solution. Mycoplasma contamination was controlled using the Jena Biosciences kit according to the manufacturer’s instructions.

### 4.4. siRNA Knockdown

To test the hypothesis about the involvement of different Gα subunits in the implementation of the cytotoxic and pro-proliferative effects of the GPR55 receptor, the knockdown approach of the Gα_12_, Gα_13_, and Gα_q_ subunits using siRNA was used. For each gene, a commercially available combination of three siRNAs was used, and transfection of a commercially available siRNA with a random nucleotide sequence (scrambled) was used as a control.

To optimize the conditions, two variants of transfecting agents were used (Thermo RNAiMax and Thermo Lipofectamine 3000, the latter in two addition variants, 0.75 and 1.5 µL per well of a 24-well plate, respectively). For Gα_13_, the use of RNAiMax turned out to be optimal; for the remaining two Gα, Lipofectamine 3000 in variant 20 was optimal. Under optimal conditions, RNA expression of each of the subunit’s gene was not recorded after 72 h of incubation with siRNA.

### 4.5. CRISPR Knockdown

To obtain *Gpr55* knockout cells into the PX459 vector at the BbsI restriction sites, oligonucleotides were cloned that were complementary to the modified Gpr55fw gene region, 5′-CACCGTCCCTATCTACAGTTTCCAT-3′ and Gpr55rev, 5′-AAACATGGAAACTGTAGATAGGGAC-3′. The genomic DNA contained the NcoI restriction site at the Cas9 cleavage site. Sequencing of the resulting plasmid confirmed the presence of the target insert.

Next, the MDA-MB-231 cell line was transfected with the obtained plasmid DNA using Lipofectamine 3000 (Invitrogen) according to the manufacturer’s protocol. Two days after transfection, cells were selected on puromycin, an antibiotic was added to the cells at a concentration of 1 to 3 μg/mL, and the cells were incubated for three days. Non-transfected MDA-MB-231 cells were used as controls. After selection on puromycin, genomic DNA was isolated from the cells, and the gene region containing the target modifiable sequence was amplified with primers hGpr55_check_fw, 5′-GGTGGAGTGCCTTTTACTTCGTCAGC-3′, and hGpr55_check_rev, 5′-TCTGCTGCACCCAGTCCTGGGTGTG-3′.

Individual cell clones were then obtained by multiple dilution and genomic DNA was isolated from the obtained clones. The PCR fragment containing the modified gene region was cloned into the pAL2-T vector.

To obtain *Cnr1* and *Cnr2* knockout cells, oligonucleotides were cloned into the PX459 vector at the BbsI restriction sites, complementary to the regions of the modifiable cnr1rev genes, 5′-AAACGCAGCGGAGGCTGCGGGAGTC-3′ and cnr1fw, 5′-CACCGACTCCCCGCAGCCTCCGCTGC-3′. For cnr2, primers cnr2rev, 5′-AAACAGGGTCACCAGTGCCCTTCCC-3′, and cnr2fw, 5′-CACCGGGAAGGGCACTGGTGACCCT-3′, were used. For knockout detection, primers test cnr1_fw, 5′-GAGAACTTCATGGACATAGAGTG-3′, and test cnr1_rev, 5′-GGAGGCCGTGACCCCACCCAGTT-3′, for gene cnr1, cnr2test_fw, 5′-TGTCTTCCTGCTGAAGATTGGCAG-3′, and cnr2test_rev, 5′-CGGAAAAGAGGAAGGCGATGAACAG-3′.

To obtain *Gpr18* knockout cells, oligonucleotides were cloned into the PX459 vector at the BbsI restriction sites, complementary to the region of the modified gene Gpr18_fw—5′-CACCGGGCGTACTTCGGCTGTACAA-3′ and Gpr18_rev—5′-AAACTTGTACAGCCGAAGTACGCCC-3′. After selection on puromycin, genomic DNA was isolated from the cells, and the gene region containing the target modifiable sequence was amplified with primers Gpr18_test_fw—5′-GTGGCATTAGTGGACTTGATATT-3′ and Gpr18_test_rev—5′-CAGTCGAGTGAGGTTCAGCAC-3′.

### 4.6. RT-qPCR

Real-time PCR with gene-specific primers was used to control the effectiveness of knockdown:

*Gα_12_* forward 5′-GGGCGAGTGAAACTGAAA-3′; reverse 5′-CACAACACGGTCCTCAATTAAAC-3′.

*Gα_13_* forward 5′-CACTGCTTAAGAGACGTCCAA-3′; reverse 5′-CAGTGGTGAAGTGGTGGTATAA-3′.

*Gα_q_* forward 5′-GCCACAGACACCGAGAATATC-3′; reverse 5′-GGTGTCTAGGAGGCACAATTAG-3′.

Beta-2 microglobulin forward 5′-CAGCAAGGACTGGTCTTTCTAT-3′; reverse 5′-ACATGTCTCGATCCCACTTAAC-3′.

For each pair of primers, it was mandatory to check nonspecific amplification using RNA without adding reverse transcriptase at the stage of cDNA production. The SYBR Green HS 2x master mix was used to perform the analysis. The primer concentration was 0.5 µM. The amplification protocol was as follows: 95 °C for 2 min, cyclic 95 °C for 10 s, 57 °C for 20 s, 72 °C for 15 s for 40 cycles; a Bio-Rad C1000 thermal cycler (Bio-Rad, Hercules, CA, USA) was used. After the amplification, PCR product melting curve was recorded in the range from 65 to 95 °C.

### 4.7. RNA Isolation and RT-PCR

Total RNA was isolated using the Total RNA Purification Kit (Jena Biosciences) according to the manufacturer’s protocol. Residual genomic DNA was removed using DNase I (Thermo Fisher Scientific, Walthon, MA USA) according to the manufacturer’s protocol; 1 U of the enzyme was used per RNA sample. cDNA was synthesized using the MMLV reverse transcription kit (Evrogen, Moscow, Russia) with an oligo-dT primer. PCR was performed using the DreamTaq master mix (Thermo Fisher Scientific, Walthon, MA USA); the program was as follows. Initial denaturation at 95 °C for 3 min, cycle: denaturation at 95 °C for 30 s, annealing at 57 °C for 30 s, DNA synthesis at 72 °C for 30 s for 35 cycles, final DNA synthesis at 72 °C for 5 min. The primers were generated using the ITDNA PrimerQuest tool (https://eu.idtdna.com/PrimerQuest; accessed on 1 January 2020) and validated using the NCBI Primer-BLAST service [38].

### 4.8. Cytotoxicity and Proliferation Evaluation

Cells were seeded in 96-well plates in the amount of 15,000 per well for the cytotoxicity induction and 4000 per well for the proliferation induction in a volume of 100 µL of medium and cultured for a day. After that, the substance was added at the required concentration in the range from 0.01 to 150 μM in the form of a DMSO solution in an additional 100 μL of the medium; the final DMSO concentration was 0.5% or less. If receptor blockers were used, they were added to 50 μL of the medium one hour before the addition of the substance and then the second portion (to ensure the constancy of concentration) with the substance; the volume of the medium in which the substance was added was 50 µL in this case. For the proliferation induction studies, the medium with the delipidated FBS was used to eliminate the influence of the natural LPI. The cells were incubated with the substance for 24 h, after which the viability was determined using the MTT test. To achieve this, the culture medium in the wells was replaced with an MTT solution in Earl’s solution with the addition of 1 g/L D-glucose and incubated for 1.5 h at 37 °C under cell culture conditions. After that, an equal volume of 0.04 M HCl in isopropanol was added to the wells and intensively stirred at 37 °C on a shaker for 30 min. In the end, the optical density of the solution was determined at a wavelength of 570 nm with a subtraction of the background at a wavelength of 660 nm. Each experiment was repeated at least five times.

### 4.9. Western Blot

To evaluate the expression of particular proteins in the cells, the cells were seeded at the density of 200,000 per well of a 24-well plate the day before the experiment. After the appropriate treatment, the cells were washed once with PBS, lysed using the lysis solution (150 mM NaCl, 1% Triton X-100, 0.1% SDS, 50 mM Tris-HCl pH 8.0, 1% protease inhibitor cocktail) for 30 min at +4 °C, and centrifuged for 5 min at 10,000× *g*. The total protein concentration in the supernatants was determined using the BCA assay. Proteins were separated using denaturing SDS-PAGE in 10% gel with the PageRuler protein ladder, transferred to a nitrocellulose membrane using the Invitrogen Power Blotter with the Invitrogen Power Blotter 1-step transfer buffer and Invitrogen precut membranes and filters, and stained with antibodies using the Invitrogen iBind system according to the manufacturer’s protocol. The following antibodies were used: rabbit anti-GPR55 (Abcam ab203663), rabbit anti-CB2 (Abcam ab45942), rabbit anti-CB1 (Abcam ab23703) mouse anti-beta-actin (Abcam ab8226); secondary antibodies (coupled to alkaline phosphatase) anti-rabbit IgG (Sigma-Aldrich A9919), anti-mouse IgG (Santa-Cruz Biotech scbt-2008). After the staining, the membrane was washed in H_2_O for 10 min and incubated with the staining solution (20 µL BCIP solution + 30 µL NBT solution per 10 mL of substrate buffer) for 1 h at room temperature. Substrate buffer for alkaline phosphatase: 100 mM Tris-HCl, pH 9.5, 100 mM NaCl, and 5 mM MgCl_2_. BCIP solution: 20 mg/mL 5-Bromo-4-chloro-3-indolyl phosphate-toluidine (BCIP) in 100% dimethyl formamide. NBT staining solution: 50 mg/mL nitro blue tetrazolium (NBT) in 70% dimethyl formamide.

### 4.10. BCA Protein Assay

Protein concentration was determined using the BCA assay [39]. The following base reagents were used: Reagent A (bicinchoninic acid 1%, Na_2_CO_3_*H_2_O 2%, sodium tartrate 0.16%, NaOH 0.4%, NaHCO_3_ 0.95%, pH 11.25), Reagent B (4% CuSO_4_*5H_2_O), S-WR (50 volumes of Reagent A + 1 volume of Reagent B). A total of 5 µL of cell lysate was mixed with 40 µL of S-WR and incubated for 15 min at 60 °C, after which the optical density was measured at λ = 562 nm using the Hidex Sense Beta Plus microplate reader (Hidex, Finland). Each sample was assayed in triplicate. Cell lysis buffer was used as a background control. Bovine serum albumin solution in the cell lysis buffer was used as a positive control and to build a calibration curve.

### 4.11. BrdU Proliferation Assay

The stimulation of the cell proliferation was validated using the BrdU cell proliferation kit (Abcam, Cambridge, MA, USA). The cells were seeded in 96-well plates at the density of 4000 per well and grown overnight. After that, DHA-DA solution in the fresh culture medium with the delipidated serum was added to the cells with full medium replacement. On day 3, BrdU reagent was added to the cells for 24 h, and the assay was completed according to the manufacturer’s protocol.

### 4.12. Caspase Activation Assay

The determination of caspase activity was performed using the specific substrates with a fluorescent 7-amido-4-trifluoromethylcoumarin (AFC) label. Cells were seeded into a 96-well plate (7 × 10^4^ cells/well) and incubated overnight. The test compounds solutions in the full culture medium were added to the cells without medium change and incubated in a CO_2_ incubator for 4 h at 37 °C. A pan-caspase inhibitor Z-VAD-FMK (100 μM) was used as a negative control. Then, the medium was discarded and 120 μL of the caspase assay buffer (20 mM HEPES, 2 mM EDTA, 0.1% CHAPS, 5 mM dithiothreitol, protease inhibitor cocktail, pH 7.4) was added to the cells. Then, the cells were frozen at −50 °C. After thawing, 120 μL of the caspase substrates Ac-DEVD-AFC (32 μM), Ac-LEHD-AFC (32 μM), and SCP0139 (32 μM) was added to the cell lysates and incubated for 90 min at 37 °C. The released AFC determination was performed using the Hidex Sense Beta Plus microplate reader (Hidex, Turku, Finland) at λ_ex_ = 505 nm, λ_em_ = 400 nm.

### 4.13. Statistical Analysis

Statistical evaluation was performed using GraphPad Prism 9.3 software. ANOVA with the Holm–Sidak or Dunnett post-test was used to compare the obtained values; *p* ≤ 0.05 were considered significant.

## 5. Conclusions

The GPR55 receptor is a promising target for therapeutical intervention to cure some cancer types expressing this receptor. However, its practical exploitation requires a strict understanding of the mechanism of its functioning at the cellular membrane level. We found for the first time the crucial role of GPR55-CB2 interaction for the pro-proliferative activity of GPR55 agonists. This finding opens new perspectives for the design of new anti-cancer drugs with a dual action on both receptors but in an opposite way.

Based on our data, the GPR18 receptor is additionally involved in the implementation of the pro-apoptotic effect of DHA-DA, while the CB1 receptor seems to be a non-participant. When the proliferation of MDA-MB-231 cells is stimulated via the GPR55 receptor, a signal is transmitted from the GPR55 receptor to the CB2 receptor due to the formation of a heterodimer. The Gα_13_ subunit enhances the GPR55-mediated cytotoxicity but has little effect on the proliferation stimulation.

## Figures and Tables

**Figure 1 ijms-24-05524-f001:**
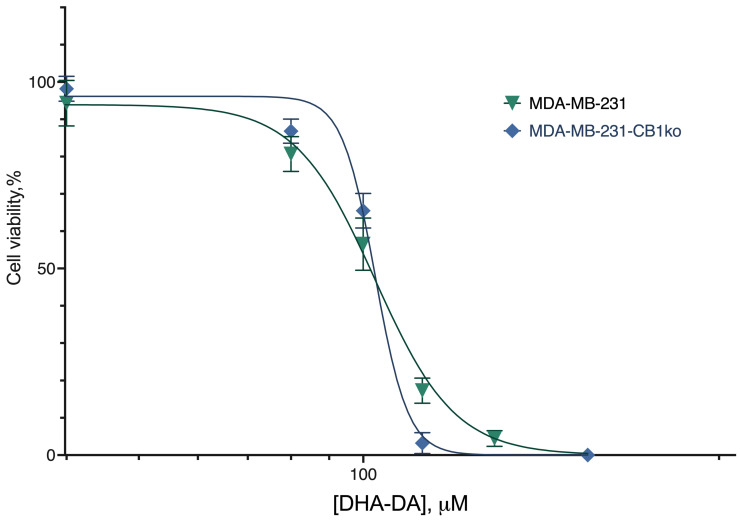
The effect of CB1 knockout on DHA-DA cytotoxicity. Incubation 24 h, MTT test, mean ± standard deviation (*n* = 7 experiments).

**Figure 2 ijms-24-05524-f002:**
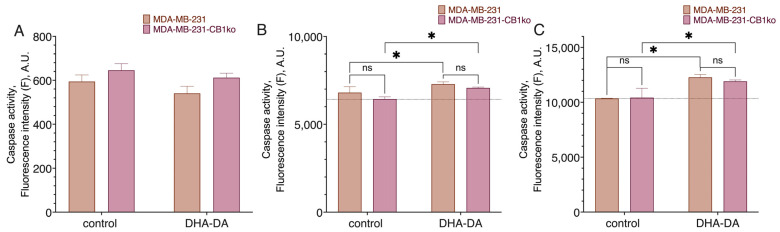
Effect of DHA-DA (40 µM) on the caspase activation in the MDA-MB-231 cell line after CB1 receptor knockout. A total of 3 h incubation time: fluorogenic substrate measurement data. (**A**) caspase 8, (**B**) caspase 9, (**C**) caspase 3; *, a statistically significant difference from the control without substance in the ANOVA test with the Holm–Sidak post-test; ns, not significant.

**Figure 3 ijms-24-05524-f003:**
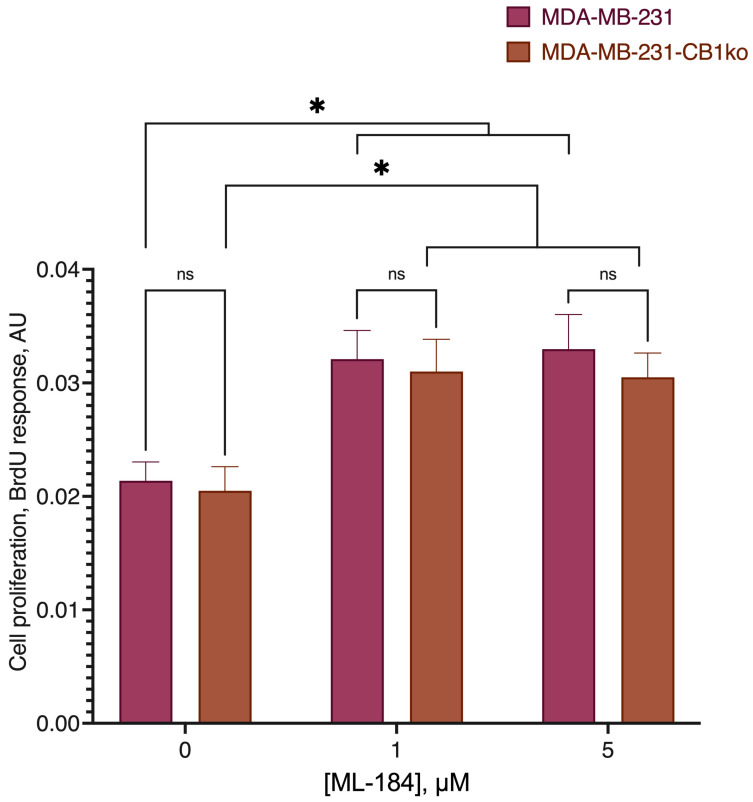
Comparison of the GPR55 agonist ML-184 activity on the original cell line and the line knocked out at the CB1 receptor. *, a statistically significant difference from the control without the substance in the ANOVA test with the Holm–Sidak post-test; ns, not significant.

**Figure 4 ijms-24-05524-f004:**
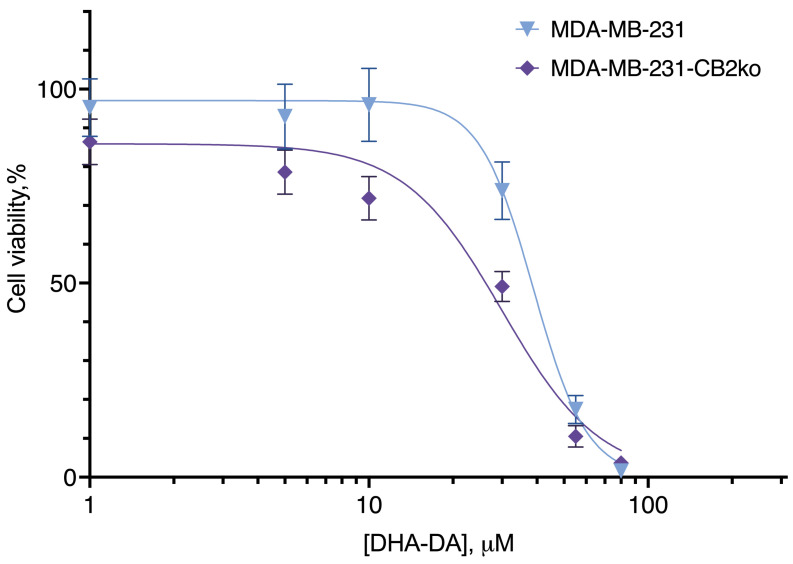
The effect of CB2 receptor knockout on DHA-DA cytotoxicity. Incubation 24 h, MTT test, mean ± standard deviation (*n* = 7 experiments).

**Figure 5 ijms-24-05524-f005:**
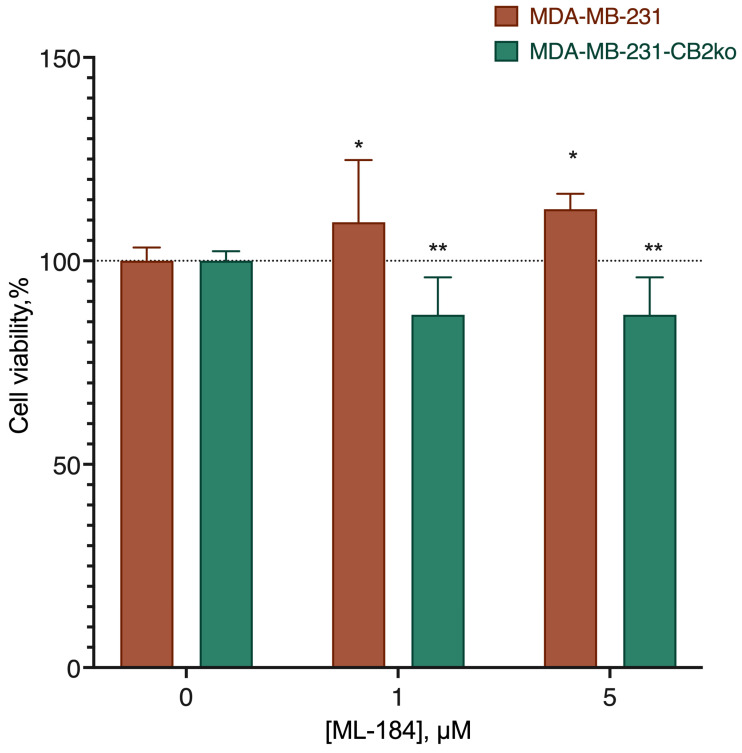
Comparison of the GPR55 agonist ML-184 activity on the original cell line and the line knocked out at the CB2 receptor. *, a statistically significant difference from the control without the substance; **, a statistically significant difference from the same concentration on the native cell line in the ANOVA test with the Holm–Sidak post-test.

**Figure 6 ijms-24-05524-f006:**
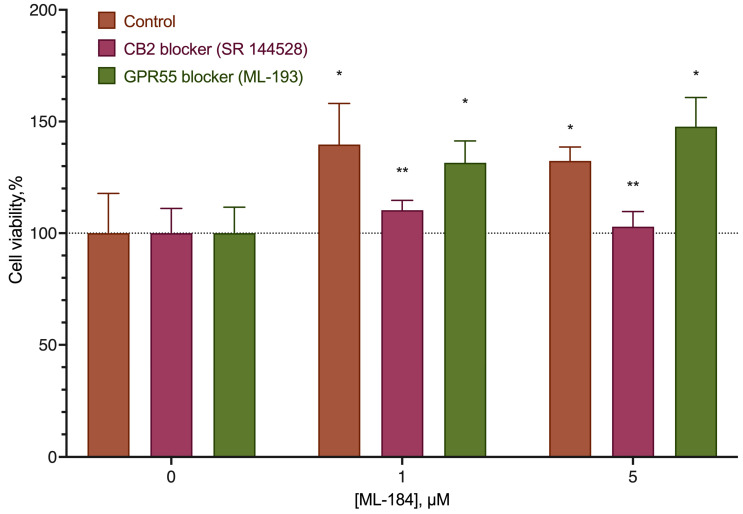
The effect of ML-184 on the original cell line against the background of GPR55 (ML-193) and CB2 (SR 144528) blockers. *, a statistically significant difference from control without ML-184; **, a statistically significant difference from the substance without blocker, and ANOVA with Holm–Sidak post-test.

**Figure 7 ijms-24-05524-f007:**
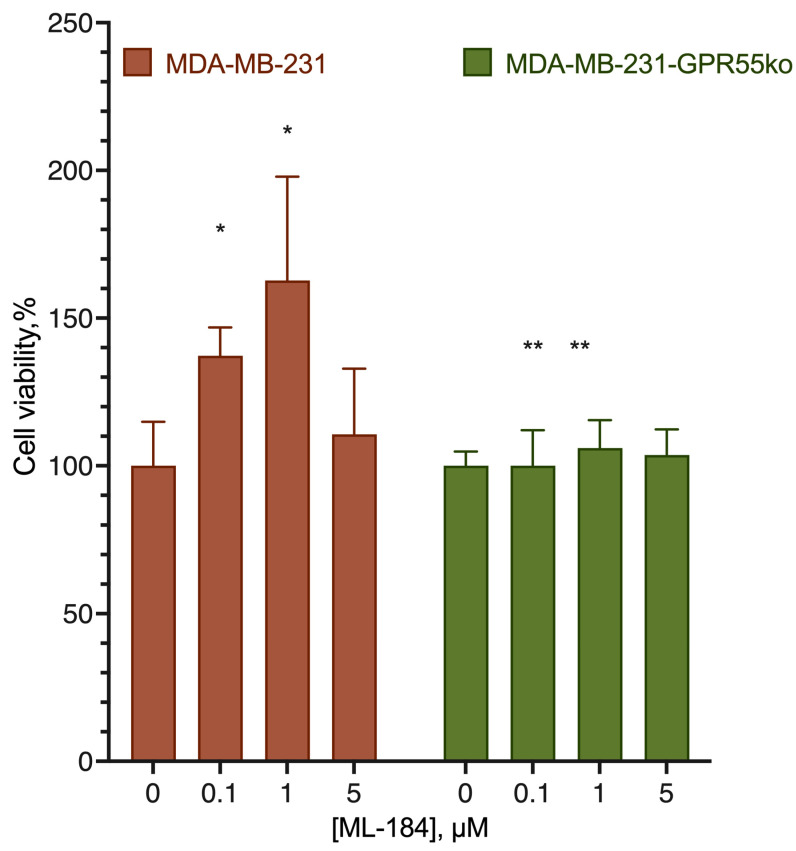
Effect of ML-184 on MDA-MB-231 cell lines knockout for the GPR55 receptor. *, a statistically significant difference from the control without substance; **, a statistically significant difference from the same concentration of the substance on the original cell line in the ANOVA test with the Holm–Sidak post-test.

**Figure 8 ijms-24-05524-f008:**
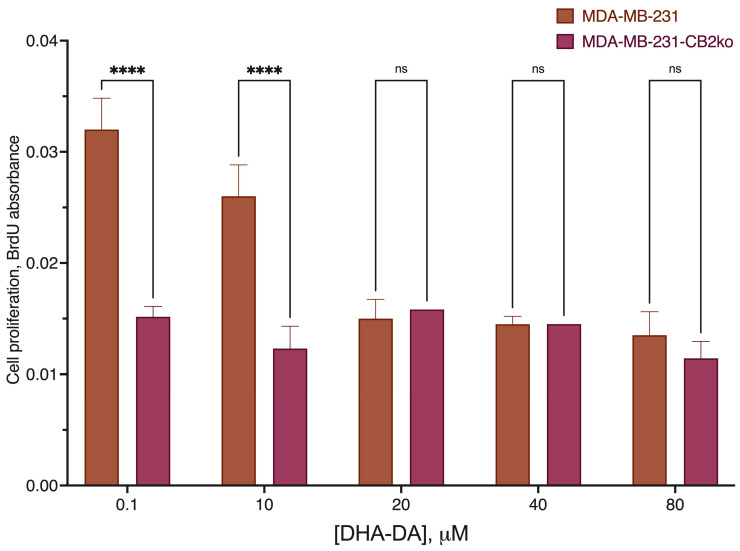
Effect of DHA-DA on the MDA-MB-231 proliferation after the CB2 receptor knockout. Delipidated serum, 72 h incubation time, BrdU proliferation kit data; ****, a statistically significant difference from the same concentration of the substance on the original cell line in the ANOVA test with the Holm–Sidak post-test; ns, not significant.

**Figure 9 ijms-24-05524-f009:**
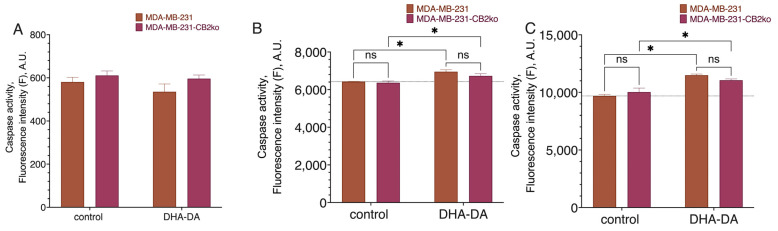
Effect of DHA-DA (40 µM) on the caspase activation in the MDA-MB-231 cell line after CB2 receptor knockout. A total of 3 h incubation time fluorogenic substrate measurement data: (**A**) caspase 8, (**B**) caspase 9, (**C**) caspase 3; *, a statistically significant difference from the control without substance in the ANOVA test with the Holm–Sidak post-test, ns, not significant.

**Figure 10 ijms-24-05524-f010:**
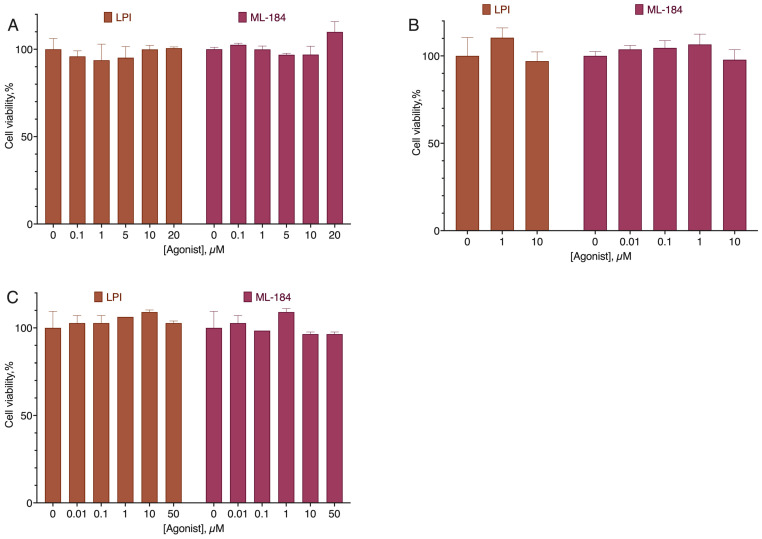
Effect of ML-184 and LPI on cell lines without the CB2 receptor expression. (**A**) Mia PaCa 2, (**B**) RPMI-8226, (**C**) Hep G2. Incubation time 72 h, MTT test data, mean ± standard error. The differences were not statistically significant. Amalgamated data of *n* = 4 experiments.

**Figure 11 ijms-24-05524-f011:**
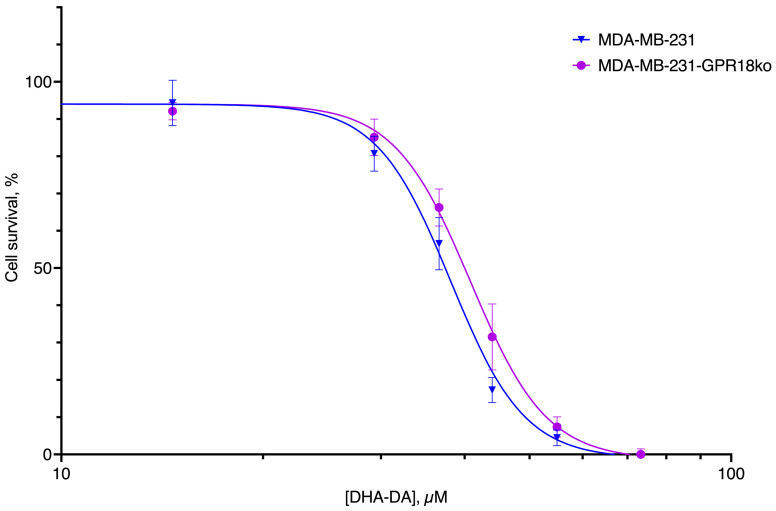
Cytotoxicity of DHA-DA for the original line MDA-MB-231 and the variant with knockout GPR18, incubation time 24 h, MTT test data. Pooled data from 7 experiments; mean ± standard error.

**Figure 12 ijms-24-05524-f012:**
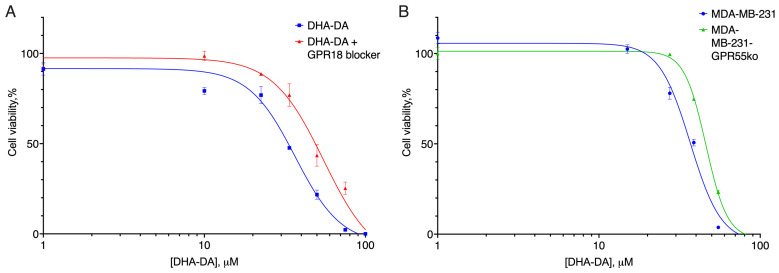
GPR18-GPR55 interaction in the DHA-DA cytotoxicity. (**A**) Cytotoxicity of DHA-DA for the original line MDA-MB-231 with and without the GPR18 blocker PSB CB5 (3 µM). (**B**) Cytotoxicity of DHA-DA for the original line MDA-MB-231 and the variants with knockout GPR55, incubation time 24 h, MTT test data. Pooled data from 7 experiments; mean ± standard error.

**Figure 13 ijms-24-05524-f013:**
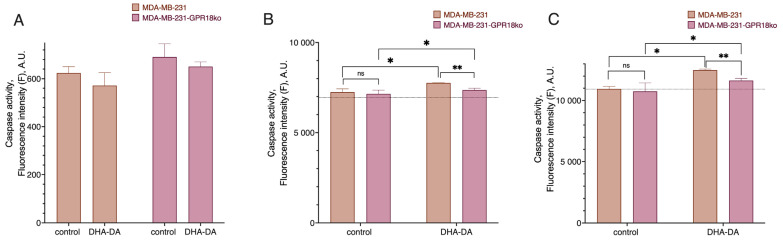
Effect of DHA-DA (40 µM) on the caspase activation in the MDA-MB-231 cell line after GPR18 receptor knockout. A total of 3 h incubation time for fluorogenic substrate measurement data. (**A**) caspase 8, (**B**) caspase 9, (**C**) caspase 3; *, a statistically significant difference from the control without substance; **, a statistically significant difference from the same treatment of the native cell line, *p* ≤ 0.05, in the ANOVA test with the Holm–Sidak post-test, ns, not significant.

**Figure 14 ijms-24-05524-f014:**
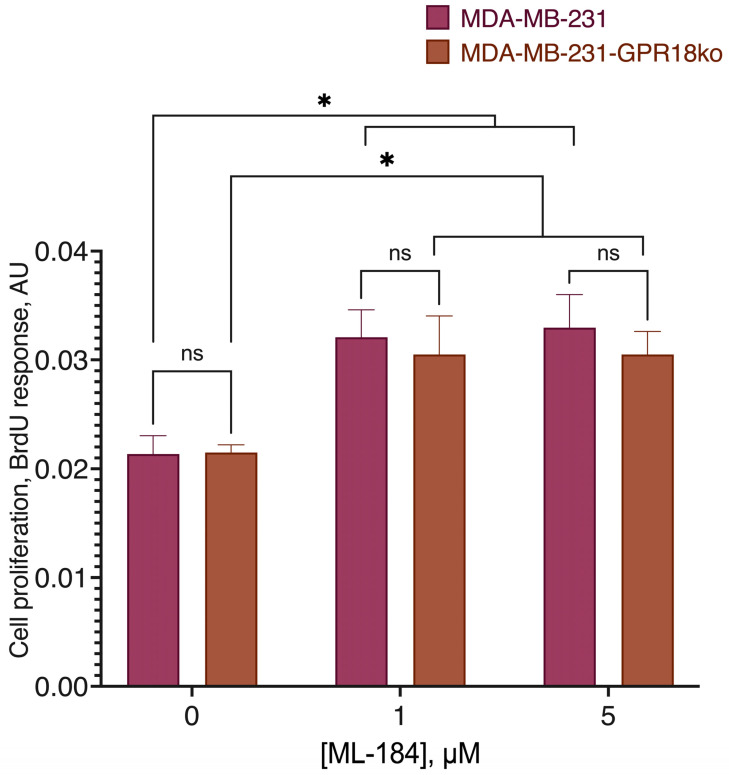
Comparison of the GPR55 agonist ML-184 activity on the original cell line and the line knocked out at the GPR18 receptor. *, a statistically significant difference from the control without the substance in the ANOVA test with the Holm–Sidak post-test, ns, not significant.

**Figure 15 ijms-24-05524-f015:**
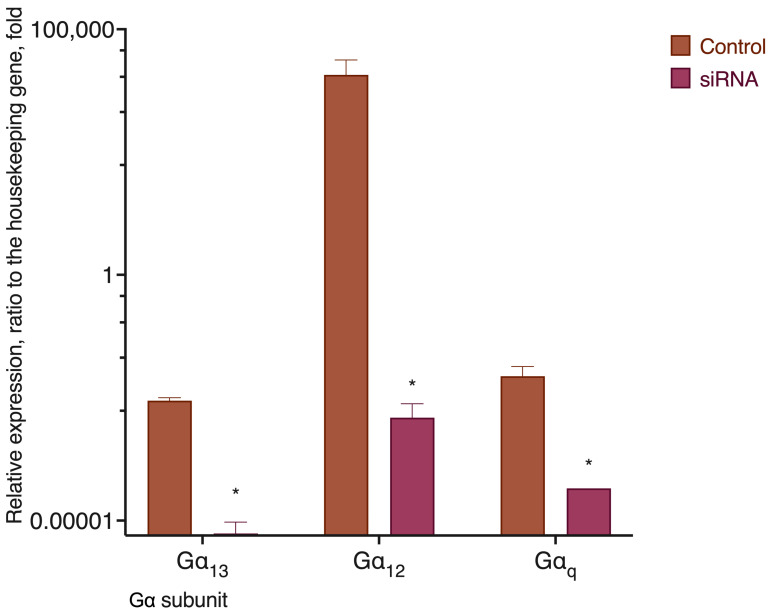
Expression of mRNA of subunits Gα_12_, Gα_13_, and Gα_q_ in MDA-MB-231 cells before and after siRNA knockdown (72 h past the transfection). RT-qPCR data, 2^(Cq(target)-Cq(reference)) ± SEM; beta-2 microglobulin gene was used as a reference. *, a statistically significant difference from the scrambled control, *p* ≤ 0.05, in the ANOVA test with the Holm–Sidak post-test.

**Figure 16 ijms-24-05524-f016:**
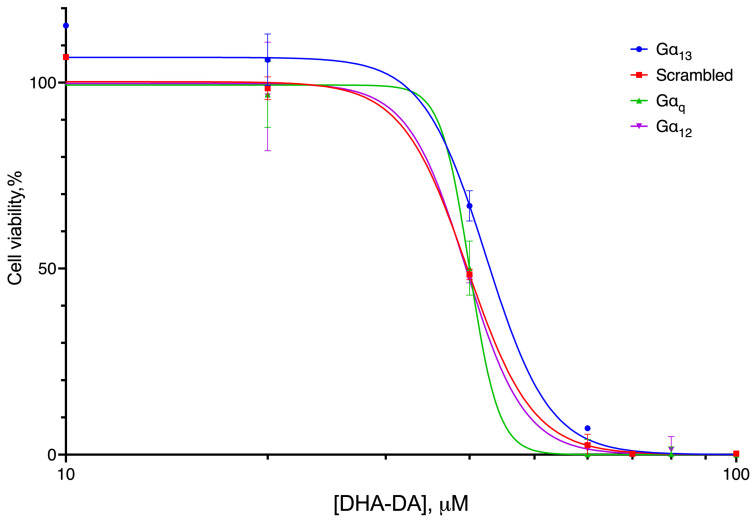
Cytotoxicity of DHA-DA for the MDA-MB-231 line with the knockdown of Gα subunits. The cells were treated for 72 h after siRNA transfection; incubation with the substance was for 24 h. MTT test data, mean ± standard error, amalgamated data of *n* = 4 experiments.

**Figure 17 ijms-24-05524-f017:**
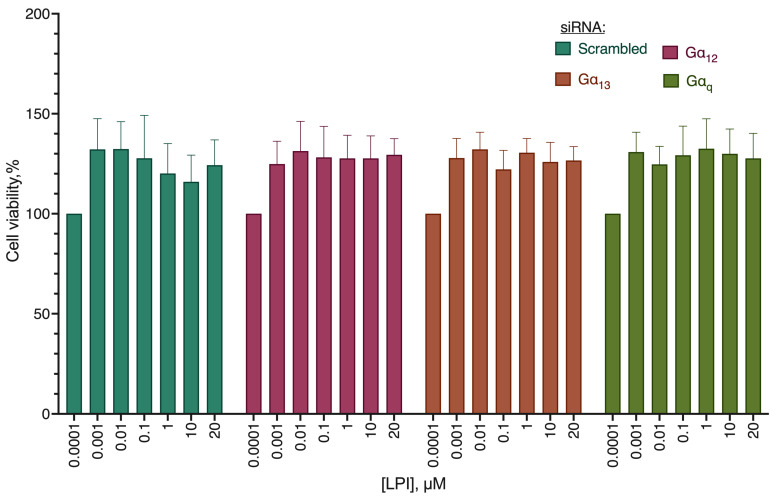
Stimulation of MDA-MB-231 proliferation against the background of knockdown of Gα subunits. Incubation with siRNA 72 h. Incubation with substance 72 h. MTT test data, mean ± standard error, amalgamated data of *n* = 4 experiments.

**Table 1 ijms-24-05524-t001:** The effect of CB1 receptor knockout on DHA-DA cytotoxicity. Incubation 24 h; MTT test.

Cell Line Variant	
Source Cell Line	CB1ko
EC_50_, µM, mean (95% CI)	
38.7 (34.62–43.0)	38.8 (33.1–41.0)

**Table 2 ijms-24-05524-t002:** The effect of CB2 receptor knockout on DHA-DA cytotoxicity. Incubation 24 h; MTT test. *, a statistically significant difference from the original cell line in the ANOVA test with the Holm–Sidak post-test.

Cell Line Variant	
MDA-MB-231	MDA-MB-231-CB2ko
EC_50_, µM, mean (95% C.I.)	
40.58 (38.13 to 43.12)	33.57 * (31.35 to 35.88)

**Table 3 ijms-24-05524-t003:** The effect of GPR18 receptor knockout on DHA-DA cytotoxicity. Incubation 24 h; MTT test.

Cell Line Variant	
MDA-MB-231	MDA-MB-231-GPR18ko
EC_50_ (95% CI), µM	
38.7 (34.62–43.0)	45.8 * (42.4–49.5)

*, a statistically significant difference from the original cell line, *p* ≤ 0.05, in the ANOVA test with the Holm–Sidak post-test.

**Table 4 ijms-24-05524-t004:** Cytotoxicity of DHA-DA for the original cell line MDA-MB-231, cells in the presence GPR18 blocker PSB CB5 (3 µM) and the variant of with knockout GPR55, incubation time 24 h, MTT test data.

Cell Line Variant		
MDA-MB-231	MDA-MB-231+PSB CB5	MDA-MB-231-GPR55ko
EC_50_ (95% CI), µM		
36.83 (34.56 to 39.18)	55.03 * (45.03 to 50.6)	46.06 * (44.75 to 47.42)

*, a statistically significant difference from the original cell line, *p* ≤ 0.05, in the ANOVA test with the Holm–Sidak post-test.

**Table 5 ijms-24-05524-t005:** Cytotoxicity of DHA-DA for the MDA-MB-231 line with the knockdown of Gα subunits. The cells were treated for 72 h after siRNA transfection; incubation with the substance was for 24 h. MTT test data.

Knockdown Variant			
Scrambled	Gα_q_	Gα_12_	Gα_13_
EC_50_ (95% CI), µM			
39.69 (39.12 to 40.19)	40.03 (39.41 to 41.05)	39.55 (38.06 to 40.55)	42.33 * (40.06 to 45.60)

*, a statistically significant difference from the original cell line, *p* ≤ 0.05, in the ANOVA test with the Dunnett post-test.

## Data Availability

The data presented in this study are available on request from the corresponding author. The data are not publicly available due to legal issues.

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
