# Peer review of "The Mechanisms of GPR55 Receptor Functional Selectivity during Apoptosis and Proliferation Regulation in Cancer Cells"

_ijms, 2023, doi:10.3390/ijms24065524_

Round 1

Reviewer 1 Report

In this manuscript Akimov and colleagues investigate pro-apoptotic signalling through GPR55 receptor. They activate the cytotoxic signal with DHA-DA. By using CRISPR-Cas9-mediated gene knockout in MDA-MB-231 breast cancer cell line authors systematically investigate the effects of cannabinoids receptors CB1, CB2, and GPR18, which potentially heterodimerize with GPR55, on signal transduction. Finally, siRNA mediated silencing of various Gα subunit is used to analyze their role as GPR55 mediators.

These are interesting results with potential implications for tumor therapy. However, there are some experimental inconsistencies, and the authors should address some issues and concerns to strengthen this paper.

Specific comments:

As the authors state, GPR55 activation can lead either to cell death via induction of apoptosis or to proliferation, depending on the ligand. The authors use an MTT assay to assess proliferation/cytotoxicity in knockout clones. However, the MTT assay (which indirectly measures the activity of the mitochondrial respiratory chain) is a good indicator of the number of viable cells, but cannot discriminate between induction of cell death or reduced proliferation, making the data difficult to be interpreted. For instance, the observed increase of DHA-DA cytotoxicity after the CB2 receptor knockout (lane 159 of the manuscript) is suggested to be a result of reduced proliferative signal in the absence of CB2 receptor, and not a result of increased apoptosis.  Although the authors utilize different stimuli to induce cytotoxicity or proliferation, the use of a more direct methods to discriminate proliferation vs cell death would be required. For instance, cell counting for pro-proliferative effects, and caspase activation/annexin V staining/tunel assay for induction of apoptosis.

It would be more informative if the experiments were more consistent. For example, authors investigate the ability of CB2 and various Gα subunit to modulate GPR55-dependent proliferative signalling (induced by either LPI or the GPR55 agonist ML184), and do not investigate the ability of CB1 and GPR18 to modulate these signals.

Lane 141 (Figure 2, Table 2). Here, it is not clear if the clones analyzed are all knockout for CB2 expression or not (lane 144 “According to the sequence analysis of clones, it was in those clones where an increase in DHA-DA cytotoxicity was observed that the expression of the CB2 receptor was disrupted”). It would be better to remove data for non-silenced clones in order to improve readability, or to keep data for only one non-silenced clone as a negative control. Same observation for Figure 3.

Lane 217. As previous, it would be better to remove data for non-silenced clones in order to improve readability. Authors write that “the target gene was modified in clone 5” but the data are shown also for clones 3, 4 and 6. I would suggest to keep data for only one non-silenced clone as a negative control.

Minor comments

Lane 185 - Figure S1 should be Figure S4

Lane 297 – “We found that this phenomenon predominantly relies on the formation of the heterodimers with the CB2 receptor”. The authors do not study the formation of heterodimers, while instead analyze a functional interaction between these receptors. Hence, I would suggest to tone down this argumentation.

Lane 300 – “Within the framework of the model system of this study, the contribution of this heterodimer was not found, as the knockdown of the CB1 receptor did not change GPR55 ligand DHA-DA cytotoxicity”.  Please rephrase the sentence or watch the punctuation, as it is not clear.

Lane 328 – “It could be supposed that the described in the literature on the heterologously expressed models agonism of LPI for GPR55 [3] is itself not always enough to induce proliferation and may require CB2 receptor presence to manifest”. Please rephrase the sentence or watch the punctuation, as it is difficult to understand.

Supplementary material – “Figure S5. The evaluation of the GPR18 receptor knockout efficiency. Sequencing results alignment with the native GPR55 mRNA” should be “native GPR18 mRNA”.

Author Response

Authors are  grateful to reviewer for valuable comments. All the necessary corrections were included within the text.

Point 1. As the authors state, GPR55 activation can lead either to cell death via induction of apoptosis or to proliferation, depending on the ligand. The authors use an MTT assay to assess proliferation/cytotoxicity in knockout clones. However, the MTT assay (which indirectly measures the activity of the mitochondrial respiratory chain) is a good indicator of the number of viable cells, but cannot discriminate between induction of cell death or reduced proliferation, making the data difficult to be interpreted. For instance, the observed increase of DHA-DA cytotoxicity after the CB2 receptor knockout (lane 159 of the manuscript) is suggested to be a result of reduced proliferative signal in the absence of CB2 receptor, and not a result of increased apoptosis.  Although the authors utilize different stimuli to induce cytotoxicity or proliferation, the use of a more direct methods to discriminate proliferation vs cell death would be required. For instance, cell counting for pro-proliferative effects, and caspase activation/annexin V staining/tunel assay for induction of apoptosis.

Response 1. BrdU proliferation data and caspase 3,9, and 8 activation data were added to the text.

Point 2. It would be more informative if the experiments were more consistent. For example, authors investigate the ability of CB2 and various Gα subunit to modulate GPR55-dependent proliferative signalling (induced by either LPI or the GPR55 agonist ML184), and do not investigate the ability of CB1 and GPR18 to modulate these signals.

Response 2. The data on ML-184 activity in CB1 and GPR18 knockouts were added.

Point 3. Lane 141 (Figure 2, Table 2). Here, it is not clear if the clones analyzed are all knockout for CB2 expression or not (lane 144 “According to the sequence analysis of clones, it was in those clones where an increase in DHA-DA cytotoxicity was observed that the expression of the CB2 receptor was disrupted”). It would be better to remove data for non-silenced clones in order to improve readability, or to keep data for only one non-silenced clone as a negative control.

Response 3. Non-silenced clones were removed.

Point 4. Same observation for Figure 3.

Response 4. Non-silenced clones removed.

Point 5. Lane 217. As previous, it would be better to remove data for non-silenced clones in order to improve readability. Authors write that “the target gene was modified in clone 5” but the data are shown also for clones 3, 4 and 6. I would suggest to keep data for only one non-silenced clone as a negative control.

Response 5. Non-silenced clones removed.

Point 6. Lane 185 - Figure S1 should be Figure S4

Response 6. Corrected

Point 7. Lane 297 – “We found that this phenomenon predominantly relies on the formation of the heterodimers with the CB2 receptor”. The authors do not study the formation of heterodimers, while instead analyze a functional interaction between these receptors. Hence, I would suggest to tone down this argumentation.

Response 7. Phrase reformulated

Point 8. Lane 300 – “Within the framework of the model system of this study, the contribution of this heterodimer was not found, as the knockdown of the CB1 receptor did not change GPR55 ligand DHA-DA cytotoxicity”.  Please rephrase the sentence or watch the punctuation, as it is not clear.

Response 8. Sentence rephrased.

Point 9. Lane 328 – “It could be supposed that the described in the literature on the heterologously expressed models agonism of LPI for GPR55 [3] is itself not always enough to induce proliferation and may require CB2 receptor presence to manifest”. Please rephrase the sentence or watch the punctuation, as it is difficult to understand.

Response 9. Sentence rephrased.

Point 10. Supplementary material – “Figure S5. The evaluation of the GPR18 receptor knockout efficiency. Sequencing results alignment with the native GPR55 mRNA” should be “native GPR18 mRNA”.

Response 10. Corrected.

Reviewer 2 Report

Journal of International Journal of Molecular Sciences

Research Article;

The article entitled “The Mechanisms of GPR55 Receptor Functional Selectivity during the Apoptosis and Proliferation Regulation in Cancer Cells’’.

The author investigate the mechanisms of multidirectional signal transduction through the same G-protein coupled receptor GPR55. Using the CRISPR-Cas9 system, clones of the MDA-MB-231 line knockout for the GPR55, CB1, CB2, and GPR18 receptor genes were obtained. On clones of the MDA-MB-231 line with a knockout CB2 receptor, the pro-apoptotic activity of the pro-apoptotic ligand docosahexaenoyl dopamine (DHA-DA) slightly increased, while the pro-proliferative activity of the most active synthetic ligand of the GPR55 receptor (ML-184) completely disappeared. In the implementation of the pro-apoptotic action of DHA-DA, the elimination of Ga13 led to a decrease of cytotoxicity. The obtained data provide novel details to the mechanism of the pro-proliferative action of GPR55.

I carefully read the manuscript and it needs critically revision for publication in the journal. I accept this article for publication after the revision. There are some common mistakes, references and English language problem in the article which should be corrected by the authors. After the correction of all the mistakes, the article could be considered for publication in the prestigious International Journal of Molecular Sciences Journal.

Comments for Authors

Ø  In abstract section “the author needs to revise the abstract with introduction, objective, method and conclusion about two or three line each”. The author focuses on result which didn’t give the ideal look to the abstract section.

Ø  Write keywords in alphabetical order.

Ø  Section Introduction; the author needs to include latest references in introduction section.

Ø  The author needs to discuss in one paragraph about DHA-DA.

Ø  Section 2-Results and Discussion, In CB2-GPR55 Interaction could the author also check the result of MTT on 48 hours?

Ø  The author needs to check the effect apoptotic effect by using flowcytometary, to see either effect of CB1 and GPR55 receptors.

Ø  Mentioned the original dimension clearly in all Figures.

Ø  Use EndNote or Mendeley software for references sequences.

Ø  Check grammatically and spelling throughout the manuscript. There are some mistakes.

Cite the following references;

v  DOI: 10.2174/1871520622666220831124321

v  doi.org/10.1038/s41419-019-2173-1

v  DOI: 10.1038/s41419-021-03771-z

v  DOI: 10.1111/jcmm.15056

Author Response

Authors are grateful to reviewer for valuable comments. All the necessary corrections were included within the text.

Point 1.  In abstract section “the author needs to revise the abstract with introduction, objective, method and conclusion about two or three line each”. The author focuses on result which didn’t give the ideal look to the abstract section.

Response 1. The journal does not require the structured abstract. The abstract reformulated to include introduction.

Point 2.  Write keywords in alphabetical order.

Response 2. Corrected.

Point 3.  Section Introduction; the author needs to include latest references in introduction section.

Response 3. References added.

Point 4.  The author needs to discuss in one paragraph about DHA-DA.

Response 4. A paragraph added to the introduction

Point 5.  Section 2-Results and Discussion, In CB2-GPR55 Interaction could the author also check the result of MTT on 48 hours?

Response 5. BrdU proliferation kit data for 72 h incubation time added.

Point 6.  The author needs to check the effect apoptotic effect by using flowcytometary, to see either effect of CB1 and GPR55 receptors.

Response 6. The pro-apoptotic effect of GPR55 activation was already published; the references are present in the introduction. The caspase activation data for the CB1, CB2, and GPR18 knockouts added.

Point 7.  Mentioned the original dimension clearly in all Figures.

Response 7. Corrected

Point 8.  Use EndNote or Mendeley software for references sequences.

Response 8. Zotero software was used.

Point 9. Check grammatically and spelling throughout the manuscript. There are some mistakes.

Response 9. Checking performed.

Point 10. Cite the following references;v  DOI: 10.2174/1871520622666220831124321; v  doi.org/10.1038/s41419-019-2173-1; v  DOI: 10.1038/s41419-021-03771-z; v  DOI: 10.1111/jcmm.15056

Response 10. The mentioned references are not relevant to the GPR55 or other cannabinoid receptor function, which it the topic of this paper

Round 2

Reviewer 1 Report

no further revision

Author Response

Point 1. no further revision
Response 1. Thank you very much once again for your in-depth evaluation.